# Learning Regularization Functionals for Inverse Problems: A Comparative Study

## Abstract

In recent years, a variety of learned regularization frameworks for solving inverse problems in imaging have emerged. These offer flexible modeling together with mathematical insights. The proposed methods differ in their architectural design and training strategies, making direct comparison challenging due to non-modular implementations. We address this gap by collecting and unifying the available code into a common framework. This unified view allows us to systematically benchmark the approaches and highlight their strengths and limitations, providing valuable insights into their future potential.

## 1   Introduction

Inverse problems are ubiquitous in imaging sciences, for example for describing the image acquisition process of magnetic resonance imaging (MRI) or computed tomography (CT). Mathematically, the reconstruction is modeled as a linear inverse problem. More precisely, we want to reconstruct an (unknown) image $\mathbf{x} \in \mathbb{R}^d$ from an observation $\mathbf{y} \in \mathbb{R}^m$ determined by the linear relation

$$\mathbf{y} = \mathbf{H}\mathbf{x} + \mathbf{n}, \tag{1}$$

where $\mathbf{H} \in \mathbb{R}^{m \times d}$ encodes the underlying data acquisition process and the noise $\mathbf{n} \in \mathbb{R}^m$ accounts for imperfections in this description. As $\mathbf{H}$ is often ill-conditioned or non-invertible, the inverse problem (1) is ill-posed in the sense of Hadamard and reconstructing $\mathbf{x}$ from $\mathbf{y}$ is challenging. A classical method to address ill-posedness is variational regularization [39], for which the unknown $\mathbf{x}$ is approximated by

$$\hat{\mathbf{x}}(\mathbf{y}) = \arg \min_{\mathbf{x}} \big\{ D(\mathbf{H}\mathbf{x}, \mathbf{y}) + \alpha R(\mathbf{x}) \big\}. \tag{2}$$

In (2), the data fidelity $D \colon \mathbb{R}^m \times \mathbb{R}^m \to \mathbb{R}$ ensures data consistency, the regularizer $R \colon \mathbb{R}^d \to \mathbb{R}$ promotes desired properties of $\mathbf{x}$, and the regularization parameter $\alpha > 0$ balances the two. The literature on variational regularization methods and their mathematical analysis is vast, see [8, 22, 39] and the references therein. In particular, the variational approach (2) leads to several desirable properties like universality, data consistency, stability and interpretability. Over the past years, deep-learning-based approaches have become the state-of-the-art for solving inverse problems [7]. However, several concerns regarding their trustworthiness remain, see, e.g., [4, 16]. In contrast, hand-crafted regularizers $R$ such as the total variation (TV) [37] are theoretically founded but cannot achieve the same reconstruction quality as data-driven approaches. We focus on the blend of these approaches, namely the learning of $R$ from data.

**Contributions**   We provide an overview of architectures and training methods for learned regularizers and provide a systematic benchmark in a unified setting. We provide implementations to all examples and highlight the individual strengths and limitations.

Submitted to 39th Conference on Neural Information Processing Systems (NeurIPS 2025). Do not distribute.

## 2 Architectures and Training Methods

**Architectures**   A pioneering learnable regularizer $R$ is the Field of Experts (FoE) [36], which is the sum of 1D potentials composed with convolutional filters. Recently, it was proposed to learn the FoE using linear splines, leading to the Convex Ridge Regularizer (CRR) [17] and Weakly Convex Ridge Regularizer (WCRR) [18]. Another convex architecture is the Input Convex Neural Network (ICNN) [7] and its descendant, namely the input weakly-convex neural network [40]. Following the idea of structured nonconvexity, these were extended to Input Difference of Convex Neural Networks (IDCNNs) [44]. Examples of more complex multiscale CNN regularizers are the Total Deep Variation (TDV) [25] and the Least Squares Residual (LSR) [46]. Exploiting the self-similarity of natural and medical images, the expected patch log likelihood (EPLL) [45] and patch normalizing flow regularizers (PatchNR) [3] define a regularizer by representing the patch distribution by a mixture model or generative model. Finally, some Plug-and-Play frameworks provably define a regularization term $R$, see [14, 21]. As an example, we inlcude Learned Proximal Networks (LPNs) [14]. Other architectures, which are not included in our comparison, are energy-based generative priors [43], dictionary learning [11, 32, 42] and priors based on generative models [2, 10, 13, 20].

**Training Methods**   First, we consider bilevel learning (BL), which adapts the $\theta$ such that the reconstruction (2) minimizes some loss. This idea started with learning only the regularization parameter $\alpha$ in (2) [12, 19, 26], and has been gradually lifted to learning regularizers. Here, we rely on Jacobian free backpropagation [9, 15] to compute the required gradients. Alternatively, the gradients can be computed by implicit differentiation [23, 30], which also allows to choose the involved step sizes and accuracies adaptively to ensure convergence [38]. A second paradigm is based on distinguishing desirable and undesirable images a priori, without actually solving (1). Prominent examples include (local) adversarial regularization (AR) [31, 34, 35] and network Tikhonov regularizers (NETT) [29]. During training, these approaches are not linked to the variational problem (2), and require the selection of a suitable regularization parameter $\alpha$ for the inverse problem at hand. Finally, from a Bayesian viewpoint, regularizers can be characterized by the log density of the training dataset. Using this identity, $R$ can be trained via maximum likelihood losses, which we use for the patch-based architectures EPLL and PatchNR. Alternatively, we can use a score matching objective. While this is computationally very efficient, we found that the results are not competitive towards BL and AR. Instead, we use it as a pretraining method to initialize $\theta$ for BL.

## 3 Comparative Study

We perform an experimental comparison of the architectures and training methods for CT reconstruction. To solve (2) efficiently, we use the nonmonotonic Accelerated Proximal Gradient algorithm [28, Supplementary material]. Our code is based on the `DeepInverse` library [41] and available online[1].

**Forward Operator and Noise Level**   We consider a sparse-view CT setting, where $\mathbf{H}$ is given by the discretized X-ray transform with 60 equispaced angles and a parallel beam geometry. We use the `DeepInverse` implementation. To keep the setup simple, we consider Gaussian noise with $\sigma = 0.7$ instead of more realistic Poisson noise.

**Datasets and Experiments**   We evaluate all models on the first test batch of the LoDoPaB-CT dataset [27] consisting out of 128 images of size $362 \times 362$ based on the LIDC/IDRI database [6]. For the training phase, we consider two different settings. In the *supervised CT* setting, we assume that we have access to training images from LoDoPaB-CT. To this end, we use 3522 images from its validation set for training. However, in practice, we rarely have training images from the same domain or scanner. Therefore, we also consider an *unsupervised CT* setting, where we train the regularizer $R$ based on natural images from the BSDS500 dataset [5, 33]. In this case, we also train on a denoising task without actually using the forward operator $\mathbf{H}$. As an error measure, we report the peak signal-to-noise ratio.

**Results**   The results for supervised CT are given in Table 1 and for unsupervised CT in Table 2. Additionally, Table 3 contains training methods that are specific to certain architectures. Moreover,

---

[1]link anonymized

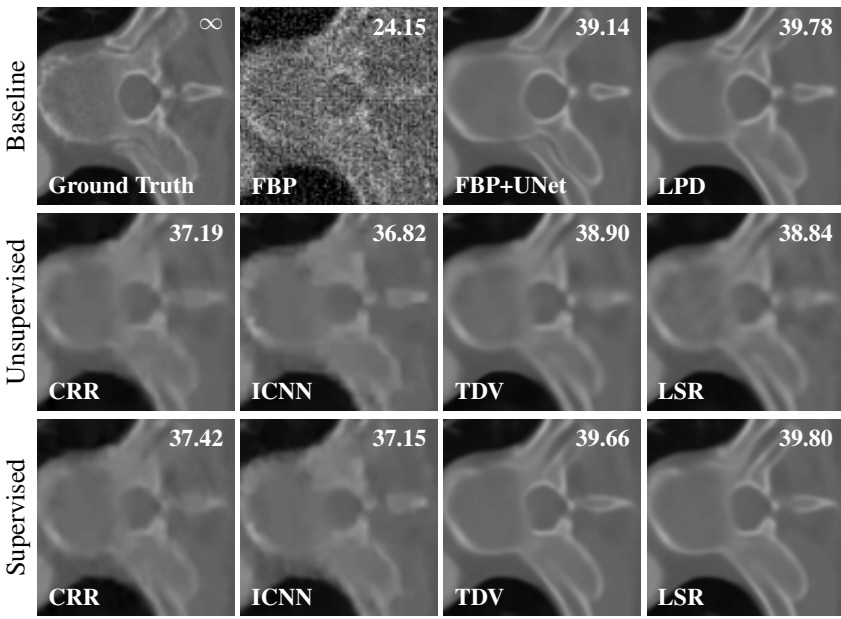

Figure 1: Reconstructions (crop) for the CT task for some baselines and BL with various architectures.

Table 1: Supervised CT experiment.

|    | CRR   | ICNN  | WCRR  | IDCNN | CNN   | TDV   | LSR   |
|----|-------|-------|-------|-------|-------|-------|-------|
| BL | 32.30 | 32.16 | 32.85 | 32.56 | -     | 33.67 | 33.72 |
| AR | 32.23 | 31.98 | 32.48 | 31.93 | 32.29 | 32.33 | -     |

Table 2: Unsupervised CT experiment.

|    | CRR   | ICNN  | WCRR  | IDCNN | CNN   | TDV   | LSR   |
|----|-------|-------|-------|-------|-------|-------|-------|
| BL | 32.17 | 31.99 | 32.65 | 32.45 | 32.69 | 33.23 | 33.11 |
| AR | 32.14 | 31.94 | 32.61 | 31.98 | 32.04 | 32.43 | -     |

Table 3: Baselines and regularizers with architecture-specific training routines.

|              | FBP   | TV    | FBP+UNET | LPD   | EPLL  | PatchNR | NETT  | LPN   |
|--------------|-------|-------|----------|-------|-------|---------|-------|-------|
| unsupervised | 19.98 | 30.99 | n/a      | n/a   | 31.94 | 32.17   | 30.64 | 31.29 |
| supervised   | n/a   | n/a   | 33.03    | 33.71 | 32.55 | 32.63   | 32.01 | 32.08 |

Table 3 includes common baselines such as the filtered back projection (FBP), the TV reconstruction, a UNet-based postprocessing of the FBP (FBP+UNet) [24] and a learned primal dual reconstruction (LPD) [1]. A hyphen indicates that the corresponding setup was unstable or required a very long training time. Visual reconstruction results for BL and some baselines are provided in Figure 1.

# 4 Discussion, Conclusions and Limitations

From our experiments, we conclude that learned regularizers like the TDV and LSR can produce competitive results to LPD, which is state-of-the-art for CT. Also the (weakly) convex architectures like (W)CRR and ICNN significantly outperform classical approaches like TV while still providing theoretical guarantees. Interestingly, the CRR consistently yields a higher PSNR than the more general ICNN architecture. Considering the training methods, BL always outperformed AR in terms of the PSNR. On the other hand, we note that AR is much faster to train, even though we do not report systematic results in this direction. Finally, all regularizers perform well in the unsupervised setting, which is a clear advantage over end-to-end reconstruction networks. Nevertheless, there is a slight drop in PSNR and also the visual impression degrades. The reconstruction times are competitive with other iterative reconstruction methods such as the well established PnP approach.

While our results give a clear impression of the capabilities and limitations of the specific architectures and training methods, several aspects remain to be explored. This includes robustness towards errors in the forward model $\mathbf{H}$, the required size of the dataset, and uncertainty estimates. Also it is unclear if training $R$ on several inverse problems can lead to a foundational model for inverse problems.

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
