# OpenReview forum: "Learning Regularization Functionals for Inverse Problems: A Comparative Study"
_EurIPS.cc/2025/Workshop/MedEurIPS — EurIPS 2025 Workshop MedEurIPS Submission_

### Official Review · Reviewer_hYSJ · 2025-10-24
**Important work, but mostly comparative.**

**Rating:** 5
**Confidence:** 4

**Review:**

Strengths:
-	Important work to have a fair comparison between methods
-	Code will be provided for a comparison framework
Weaknesses:
-	Line 68, Why is Gaussian noise simpler than Poisson noise?
-	I am not sure if this workshop is the ideal venue for this paper. As it is not a novel Idea, existing frameworks are compared. I think this fits better into a larger format with more comparison.

---

### Official Review · Reviewer_PYsN · 2025-10-31
**Nice work with experiments on CT reconstruction**

**Rating:** 8
**Confidence:** 4

**Review:**

Nice work! It's also related to medical theme as it takes CT reconstruction as experiments. I believe it will bring interesting discussions to the workshop!

---

### Official Review · Reviewer_XHYj · 2025-10-31
**Valuable Benchmark, Limited Novelty**

**Rating:** 5
**Confidence:** 4

**Review:**

This abstract presents a comprehensive benchmark of learned regularization approaches for inverse problems, unifying diverse architectures and training methods in a common framework. The effort is valuable and the paper is clearly written, but the contribution is mainly integrative rather than conceptual. The work would benefit from a deeper analysis of why certain models perform better than others and more discussion of the underlying mechanisms. Overall, this feels better suited for a larger-format paper where such insights and ablations can be fully explored. Would a comparison to more recently popular reconstruction models, e.g., deep generative priors such as diffusion models, also make sense?

---

### Decision · Program_Chairs · 2025-11-03

**Decision:**

Reject

**Comment:**

This paper presents a comprehensive benchmark of learned regularization approaches for inverse problems, including CT reconstruction experiments relevant to the medical imaging domain. While the effort to unify diverse TV methods into a common framework is valuable and the paper is clearly written (XHYj, PYsN), the consensus points toward limited conceptual novelty and insufficient depth of analysis.

Key weaknesses highlighted across the reviews include:
1. Limited conceptual contribution: The paper primarily compares existing frameworks and lacks a truly novel idea (XHYj, hYSJ).
2. Missing deep analysis: There is a lack of in-depth analysis into why certain models behave or perform better than others (XHYj).
3. Comparison with recent methods: The study would benefit from expanding comparisons to more recently popular reconstruction models, such as deep generative priors (e.g., diffusion models for inverse problems) (XHYj).

Although one reviewer found the work interesting and relevant to the medical theme, the majority sentiment is that the paper does not meet the bar for conceptual contribution required for acceptance at this workshop.